# Incidence and preventability of adverse events in adult patients admitted to a Brazilian teaching hospital

**Ariane Cristina Barboza Zanetti**[1☉]*, **Bruna Moreno Dias**[1☉], **Andrea Bernardes**[1‡], **Helaine Carneiro Capucho**[2‡], **Alexandre Pazetto Balsanelli**[3‡], **André Almeida de Moura**[1‡], **Rodrigo Soato**[1‡], **Carmen Silvia Gabriel**[1☉]

**1** Department of General and Specialized Nursing, University of São Paulo at Ribeirão Preto College of Nursing, Ribeirão Preto, São Paulo, Brazil, **2** Department of Pharmacy, Faculty of Health Sciences, University of Brasilia, Brasília, Distrito Federal, Brazil, **3** Department of Administration of Health and Nursing Services, Paulista School of Nursing, Federal University of São Paulo – UNIFESP, São Paulo – SP, Brazil

☉ These authors contributed equally to this work.
‡ These authors also contributed equally to this work.
* arizanetti@gmail.com

**Data Availability Statement:** All relevant data are within the manuscript.

**Funding:** The authors received no specific funding for this work.

## Abstract

### Objective

To analyze the incidence and preventability of adverse events related to health care in adult patients admitted to a Brazilian teaching hospital.

### Methods

A retrospective cohort study, in which the incidence and preventability of adverse events related to health care were based on a two-stage retrospective review of 368 medical records (nurses and pharmacist review of medical records, followed by physicians review of triggered medical records) of adult patients whose hospitalizations occurred during 2015 in a high-complexity public teaching hospital located in Brazil. Data were collected from February 2018 to February 2019.

### Results

A total of 266 adverse events were observed in 124 patients. The incidence of adverse events related to health care was 33.7% (95% CI 0.29–0.39), and the incidence density was 4.97 adverse events per 100 patient-days. Adverse events were responsible for 701 additional days of hospitalization, and the estimated length of additional hospital stay attributable to them was, on average, 6.8 days per event. The most common types of events were related to general care (60; 22.6%), medications (50; 18.8%), nosocomial infection (35; 13.2%), any other type (11; 4.1%), and diagnoses (2; 0.8%). Regarding the severity of adverse events, it was found that 168 (63.2%) were mild, 55 (20.7%) were moderate, and 43 (16.2%) were severe. In addition, it was estimated that 155 (58.3%) events were preventable. The length of a patient's hospital stay was identified as a risk factor for the occurrence of adverse events (RR 1.20; 95% CI 1.04–1.39).

**Competing interests:** The authors have declared that no competing interests exist.

## Conclusions

Through knowledge of the incidence, nature, severity, preventability, and risk factors for the occurrence of adverse events, it is possible to create the opportunities to prioritize the implementation of strategies for mitigating specific events based on reliable data and concrete information.

## Introduction

The risks and incidence of events that cause harm to patients have increased in all spheres of health, especially in the hospital context. Currently, the rate of adverse events (AEs) in the hospital environment is a relevant indicator for patient safety. The complex combination of processes, technologies, and human interactions, which are part of the current health care delivery system, has important benefits, but at the same time, leads to the risk of AE occurrence [1–4].

Starting with the 1990 Harvard Medical Practice Study (HMPS) and then the 1999 report by the Institute of Medicine (IOM), entitled "To err is Human", discussions on patient safety have been at the forefront of international debate for the last three decades. Many studies involving retrospective reviews of patient records, conducted in several countries, followed the HMPS protocol in order to assess patient safety in hospital admissions. In these studies, the incidence rates of AEs ranged from 2.9% to 16.6%, while rates of preventable AEs ranged from 1.0% to 8.6% [5–10]. These results increased the need for action to be taken to guarantee the safety of patients in hospitals. Thus, quality improvement and patient safety programs have been implemented in many countries [11].

Frequent monitoring of AE incidence and preventability rates can provide insight into the status and advances of patient safety in hospitals. Periodic and large-scale measurement of AEs is important for estimating the effect of efforts to improve patient safety. However, such measurements are complex and expensive, and it is not possible for all contextual factors that influence the occurrence of AEs to be considered [12].

The high proportions of incidence and preventability of AEs pointed out in the research emphasize the magnitude of patient safety problems in hospitals in general. The analysis of AEs provides a denser and more adequate understanding of the susceptibility to failures in healthcare systems and, once it is possible to evaluate and measure the AE that has occurred, it becomes feasible to develop strategies to improve the quality of care in health services provided by hospital institutions [13].

It is observed that the development of initiatives to protect patients from the occurrence of AEs has been limited by the lack of knowledge about the epidemiological characterization of unintentional injuries or harm that affect patients, both of which are essential to discern the types of AEs that affect patients, the probability of their occurrence, their severity, and degree of preventability [14].

Through a retrospective review of medical records, the present study addressed an important gap in the investigation related to patient safety in Brazil: the lack of epidemiological knowledge about frequency, incidence, and preventability of AEs in hospital admissions. It is important to highlight that the main Brazilian study that assessed the occurrence of AEs in hospitals was developed in 2003 in three public hospitals in Rio de Janeiro, and the incidence and preventability rates of AEs were 8.6% and 66.7%, respectively [15].

Therefore, the objective of this study was to analyze the incidence and preventability of AEs related to health care in adult patients admitted to a Brazilian teaching hospital.

## Materials and methods

This study followed a retrospective cohort study design, in which the incidence and preventability of AEs related to health care were based on the retrospective review of medical records of adult patients whose hospitalizations occurred during 2015 in a high-complexity public teaching hospital, the General Hospital of the Medical School of Ribeirão Preto of the University of São Paulo, located in the state of São Paulo, Brazil.

In order to allow for comparisons between studies, incidence and preventability were measured through the application of a protocol developed by the researchers responsible for the Iberoamerican Study of Adverse Events (IBEAS) [16], which had as a precursor the protocol developed by the HMPS. In addition, we used the method of retrospective review of medical records proposed by researchers from the Canadian Adverse Event Study (CAES) [6], one of the most relevant derivations of the HMPS method.

Adverse events were defined as any unexpected occurrence, evidenced in the medical record, which caused harm to the patient (injury, disability, prolonged hospital stay, sequelae at discharge, and/or death). AEs were considered to be those related to the health care provided to the patients, and not to the evolution of their disease, being classified according to severity into mild (there was no prolongation of hospital stay); moderate (there was an extension of the stay by at least one day); and severe (death, disability at discharge, or surgical intervention required) [16].

Each patient could have one or more AEs evaluated according to the relationship with health care and causality. Regarding the time of occurrence and identification of AEs, they could happen: in the index hospitalization (admission of interest); between hospital admission and discharge or death; as a result of previous admission (within the preceding 12 months) to the same hospital; or as a consequence of index hospitalization causing reentry (over the following 12 months) in the same hospital [6, 16]. AEs were considered to be preventable when there was evidence that they could have been prevented by adopting managerial and/or care practices different from those applied [17].

The study population consisted of adult patients admitted to the hospital in 2015, aged at least 18 years, hospitalization for more than 24 hours of hospital stay (or death in less than 24 hours), admission to several units and/or medical specialties, except psychiatry and obstetrics. Patients whose medical records were unavailable at the time of data collection and/or follow-up by the palliative care team were excluded. A total of 10358 index admissions were eligible for inclusion in a random sample. The parameters used for determining the sample size were the probability of occurrence of AEs of 8.6%, significance level of 5%, absolute error of 3%, and an estimated loss of 10%. The final sample size was 370 inpatient admissions, with 368 eligible for the study.

A two-stage retrospective medical records review was carried out, and the data were collected from February 2018 to February 2019. The first stage aimed to identify the demographic, clinical, and hospitalization profiles of patients and the presence of a potential AE (pAE). This phase was based on the explicit review of medical records by five nurses and one pharmacist, using 19 screening criteria for the presence of pAEs stipulated by the instrument. The presence of at least one of the 19 criteria denoted the eligibility of the medical record for the second stage of the review. The second stage evaluated the occurrence of AEs related to health care and their preventability through an implicit structured review carried out by three physicians. Thus, from the pAEs identified in the screening phase, physicians corroborated or refuted the evidence of occurrence of AEs.

Based on professional judgment, physicians assessed the relationship between AEs and the health care provided using a six-level scale that indicated the accuracy of this relationship

(1-No evidence that care is the cause of AE; 2-Minimal likelihood that care is the cause; 3-Slight likelihood that care is the cause; 4-Moderate likelihood that care is the cause; 5-Very likely that care is the cause; 6-Total evidence that care is the cause). The same method was used to assess preventability (1-No evidence of preventability; 2-Minimal possibility of preventability; 3-Slight possibility of preventability; 4-Moderate possibility of preventability; 5-High possibility of preventability; 6-Total evidence of preventability). Cases in which the score was ≥4 points were considered as AEs related to health care and as preventable. It is important to mention that both scales were previously adapted to the context of Brazilian hospitals [15]. Physicians also estimated the length of hospital stay directly attributable to AEs, the need for additional diagnoses and/or treatments, the classification of AEs, and their severity (mild, moderate, severe).

Statistical analyses were performed using IBM SPSS (version 25.0.), R i386 (version 3.4.0.), and statistical analysis system (SAS). Descriptive statistics were used for all study variables to characterize the sample. Categorical variables were presented using absolute and relative frequencies, while numerical variables were represented by mean, standard deviation, median, minimum, and maximum values. The Mann-Whitney U test for independent samples, Fisher's exact test, and Pearson's chi-square test were used to perform inferential statistical analysis (significance set at $p < 0.05$). Log-binomial regression analysis was used to estimate the risk factors associated with the occurrence of adverse events. The results of the regression analysis were presented as relative risk (RR) and 95% confidence intervals (CIs). The levels of statistical significance of the variables were determined at $p < 0.05$. Other parameters of interest in this study were the incidence of AEs, incidence density of AEs per 100 patient-days, and proportion of preventable AEs.

Ethics approval was obtained from the Research Ethics Committee of the University of São Paulo at Ribeirão Preto College of Nursing.

## Results

The subjects who comprised the sample of this study were predominantly female, over 55 years old, white, and had completed their elementary education. Elective hospitalizations and hospital discharge were highlighted, and the average length of stay was 6.8 days.

pAEs were identified in 166 (45.1%) of the 368 patients sampled. Among the subjects with pAEs, 149 patients had a total of 386 incidents, with or without harm. The stages of retrospective review of medical records are detailed in Fig 1.

Therefore, 266 AEs were observed in 124 patients, and the incidence of AEs related to the health care provided was 33.7% (95% CI 0.29–0.39), while the incidence density was 4.97 AEs per 100 patient-days.

Regarding the profile of patients who made up the second stage of retrospective review, Table 1 describes the demographic, clinical, and hospitalization characteristics of the groups of patients with and without AEs.

Amid the demographic, clinical, and hospitalization characteristics of the groups of patients with and without AEs, there was a statistically significant association for the variables average number of extrinsic factors per patient (3.9 vs. 2.7 extrinsic factors; $p = 0.021$) and average length of stay (11.4 vs. 5.7 days, $p = 0.021$) (Table 2).

It was found that 86.3% (n = 107) of patients with AEs had intrinsic risk factors, while 97.6% (n = 121) had extrinsic risk factors. The intrinsic risk factors with the highest number of occurrences were arterial hypertension (n = 65; 52.4%), diabetes (n = 42; 33.9%), neoplasia (n = 32; 25.8%), and hypercholesterolemia (n = 27; 21.8%). The most frequent extrinsic risk factors in hospitalizations were the presence of a peripheral venous catheter (n = 120; 96.8%),

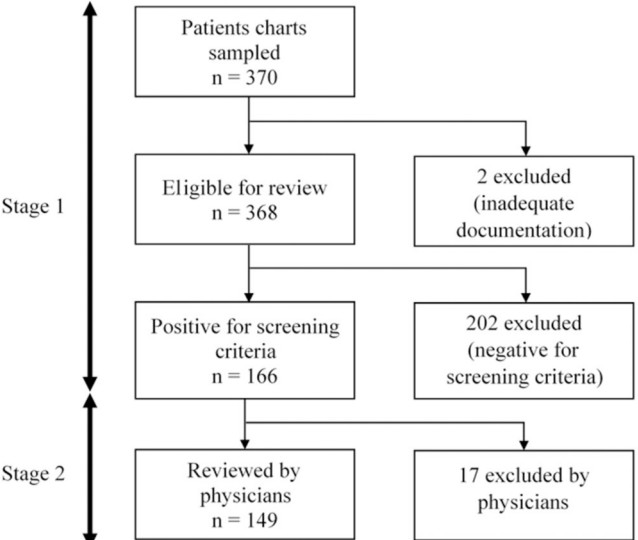

**Fig 1. Flowchart of the process of retrospective review of medical records.**

tracheal intubation and mechanical ventilation (n = 63; 50.8%), use of a continuous infusion pump (n = 47; 46.0%), and the presence of an indwelling urinary catheter (n = 54; 43.5%).

Among patients with AEs, 69 (55.6%) had more than one event, with an average of 2.1 AEs per patient. The AEs were responsible for 701 additional days of hospitalization, and the estimated length of additional hospital stay attributable to the AEs was, on average, 6.8 days per event.

In view of the clinical context of the patients, 44.7% (n = 119) of AEs were unlikely to occur and 63.5% (n = 169) of AEs did not cause an increase in the length of stay. However, 31.6% (n = 84) and 66.2% (n = 176) required additional diagnostic tests and treatments, respectively.

The classification of AEs according to nature, preventability, and severity as well as the presence of comorbidities and the hospitalization sector in which they occurred, are shown in Table 3. There was a statistically significant association between the severity categories of the events and the variable classification of AEs according to their nature ($p <0.001$) and hospitalization sector ($p = 0.002$).

Among the 155 preventable AEs, in 54 cases there was an increase in the length of hospital stay, cumulatively responsible for 377 additional days of hospitalization. Regarding these preventable AEs that led to additional days of hospitalization, the estimated length of additional hospital stay was approximately seven days per event. In addition, a statistically significant association was found ($p = 0.031$) between the preventability and classification of AEs according to their nature, with emphasis on events related to procedures, whose percentage of preventability was predominant, as shown in Table 4.

Table 5 shows that the length of hospital stay was identified as a risk factor for the occurrence of AEs, since a length of stay of eight days or more increased the risk of occurrence of AEs by 1.2 times (20%) compared to hospitalizations of less than eight days (RR 1.20; 95% CI 1.04–1.39).

## Discussion

Based on the results of the retrospective review of medical records, it was observed that, among the 166 patients with positive screening for pAEs, 17 (10.2%) were false-positive. Other

**Table 1. Demographic, clinical, and hospitalization characteristics of the groups of patients with and without adverse events.**

| Variables | Adverse event | | | | p-value |
|---|---|---|---|---|---|
| | Yes | | No | | |
| | n | % | n | % | |
| **Demographic variables** | | | | | |
| **Sex** | | | | | |
| Female | 62 | 50.0 | 15 | 60.0 | 0.361[a] |
| Male | 62 | 50.0 | 10 | 40.0 | |
| **Age group** | | | | | |
| <60 years old | 65 | 52.4 | 13 | 52.0 | 0.969[a] |
| 60 years or older | 59 | 47.6 | 12 | 48.0 | |
| **Education** | | | | | |
| None | 12 | 9.7 | 2 | 8.0 | 0.656[b] |
| Incomplete elementary school | 21 | 16.9 | 3 | 12.0 | |
| Complete elementary education | 57 | 46.0 | 14 | 56.0 | |
| Incomplete high school | 3 | 2.4 | 2 | 8.0 | |
| Complete high school | 21 | 16.9 | 3 | 12.0 | |
| Complete higher education | 10 | 8.1 | 1 | 4.0 | |
| **Race/ethnicity** | | | | | |
| White | 105 | 84.7 | 22 | 88.0 | - |
| Black | 7 | 5.6 | 1 | 4.0 | |
| Mixed | 11 | 8.9 | 2 | 8.0 | |
| Asian | 1 | 0.8 | 0 | 0.0 | |
| **Clinical variables** | | | | | |
| **Comorbidity** | | | | | |
| No | 3 | 2.4 | 1 | 4.0 | 0.524[b] |
| Yes | 121 | 97.6 | 24 | 96.0 | |
| **Intrinsic factors** | | | | | |
| No | 17 | 13.7 | 3 | 12.0 | 1.000[b] |
| Yes | 107 | 86.3 | 22 | 88.0 | |
| **Extrinsic factors** | | | | | |
| No | 3 | 2.4 | 1 | 4.0 | 0.524[b] |
| Yes | 121 | 97.6 | 24 | 96.0 | |
| **Procedure** | | | | | |
| No | 33 | 26.6 | 10 | 40.0 | 0.178[a] |
| Yes | 91 | 73.4 | 15 | 60.0 | |
| **Prognosis—Complete recovery** | | | | | |
| No | 29 | 23.4 | 9 | 36.0 | 0.187[a] |
| Yes | 95 | 76.6 | 16 | 64.0 | |
| **Prognosis—Residual disability** | | | | | |
| No | 98 | 79.0 | 16 | 64.0 | 0.106[a] |
| Yes | 26 | 21.0 | 9 | 36.0 | |
| **Prognosis—Terminal illness** | | | | | |
| No | 121 | 97.6 | 25 | 100.0 | 1.000[b] |
| Yes | 3 | 2.4 | 0 | 0.0 | |
| **Hospitalization variables** | | | | | |
| **Hospital admission** | | | | | |

(*Continued*)

**Table 1.** (Continued)

| Variables | Adverse event | | | | p-value |
|---|---|---|---|---|---|
| | Yes | | No | | |
| | n | % | n | % | |
| Elective | 104 | 83.9 | 21 | 84.0 | 1.000[b] |
| Urgency | 20 | 16.1 | 4 | 16.0 | |
| **Hospital discharge** | | | | | |
| Discharged alive | 113 | 91.1 | 23 | 92.0 | 1.000[b] |
| Died | 11 | 8.9 | 2 | 8.0 | |
| **Hospitalization sector** | | | | | |
| Clinical | 47 | 37.9 | 12 | 48.0 | 0.626[b] |
| Surgical | 73 | 58.9 | 13 | 52.0 | |
| Intensive / semi-intensive care | 4 | 3.2 | 0 | 0.0 | |
| **Total** | 124 | 100 | 25 | 100 | - |

p-values were calculated using
[a]Pearson's chi-square test and
[b]Fisher's exact test.

authors have also reported the exclusion of patients reported as false-positive for pAEs, as described in an Irish study [18], which excluded six subjects, and in research carried out in Chile [19], which excluded 37 subjects.

Therefore, in the second stage of the present study, the reviewers analyzed the hospital records of 149 individuals, detecting 266 AEs in 124 patients. The incidence rate of AEs in hospitalized patients was 33.7%, and the incidence density was 4.97 AEs per 100 patient-days. Comparatively, Brazilian research carried out in teaching hospitals in Rio de Janeiro, with patients admitted in 2003, showed that the incidence of AEs was 8.6%, and the incidence density was 0.8 AEs per 100 patient-days [15]. A Spanish study reported an incidence of AEs of 9.3% and an incidence density of 1.2 AEs per 100 patient-days [20].

Researchers from other countries also followed similar paths to those described in this study and found the following consecutive percentages of AEs: USA (3.7%), Australia (16.6%), New Zealand (11.3%), United Kingdom (10.8%), Canada (7.5%), Denmark (9%), France (14.5%), Sweden (12.3%), Tunisia (10%), Netherlands (5.7%), Italy (5.2%), Portugal (11.1%), Colombia (4.5%), Mediterranean countries (8.2%), Latin American countries (10.5%), Iran (7.3%), Belgium (7.1%), Ireland (10.3%), and Switzerland (14.1%) [5–10, 17, 21–31]. It is noteworthy that all of the cited studies showed variations in the method of retrospective review, since there were different screening criteria, heterogeneity in the training of reviewers, plurality between definitions, diversity in periods and places of data collection, and differences in causality and preventability assessments [32].

It is essential to interpret the superiority verified in the incidence value estimated by this work in relation to other studies of international and even national scope. Most of the studies published on this theme analyzed only the AEs considered to be of greater severity for each patient. In contrast, the present study recorded all AEs experienced by patients, which enabled a more detailed incidence estimate. The value of the incidence rate may vary depending, for example, on the population of patients sampled, place of study, organizational safety culture of the hospital, concept and threshold of causation of AEs, extent of the revised documentation, time of occurrence of AEs in relation to index admission, and other issues related to the operationalization of the research [18].

**Table 2. Demographic, clinical and hospitalization variables of the groups of patients with and without adverse events.**

| Variables | Adverse event | | p-value[a] |
|---|---|---|---|
| | Yes (n = 124) | No (n = 25) | |
| **Demographic variables** | | | |
| **Age** | | | |
| Average | 56.5 | 58.5 | 0.606 |
| Median | 59.1 | 59.0 | |
| Standard deviation | 17.1 | 17.3 | |
| Minimum | 18.1 | 25.4 | |
| Maximum | 87.4 | 87.6 | |
| **Clinical variables** | | | |
| **Comorbidities** | | | |
| Average | 5.8 | 5.5 | 0.781 |
| Median | 6.0 | 6.0 | |
| Standard deviation | 3.0 | 2.8 | |
| Minimum | 0.0 | 0.0 | |
| Maximum | 15.0 | 10.0 | |
| **Intrinsic factors** | | | |
| Average | 2.8 | 2.7 | 0.823 |
| Median | 2.5 | 2.0 | |
| Standard deviation | 2.2 | 2.0 | |
| Minimum | 0.0 | 0.0 | |
| Maximum | 9.0 | 7.0 | |
| **Extrinsic factors** | | | |
| Average | 3.9 | 2.7 | **0.021** |
| Median | 3.0 | 2.0 | |
| Standard deviation | 2.6 | 2.0 | |
| Minimum | 0.0 | 0.0 | |
| Maximum | 12.0 | 8.0 | |
| **Hospitalization variables** | | | |
| **Length of stay** | | | |
| Average | 11.4 | 5.7 | **0.021** |
| Median | 8.0 | 5.0 | |
| Standard deviation | 10.8 | 4.0 | |
| Minimum | 1.0 | 1.0 | |
| Maximum | 47.0 | 15.0 | |

[a] p-values were calculated using Mann-Whitney U test for independent samples; Significant numbers from the statistical tests are presented in bold.

We emphasize that the research hospital was a pioneer in implementing a culture of patient safety in the country, and it is observed that employees have greater freedom to report errors or AEs in medical records. Thus, it is assumed that the patient safety culture, which is very present in the study hospital, strengthens the reports of AEs in medical records and, more recently, in notification systems. We can also consider that the health team feels more comfortable reporting when its members are supported by the hospital by building a culture of safety. This fact points to the need to study the relationship between culture and the reporting of AEs in medical records or notifications.

**Table 3. Associations between the severity categories of adverse events and the variables classification of the events according to their nature, preventability, comorbidity and hospitalization sector.**

| Variables | Severity of AE | | | | | | Total | | p-value |
|---|---|---|---|---|---|---|---|---|---|
| | Mild | | Moderate | | Severe | | | | |
| | n | % | n | % | n | % | n | % | |
| **Classification of AE** | | | | | | | | | |
| Related to general care | 52 | 31.0 | 6 | 10.9 | 2 | 4.7 | 60 | 22.6 | <0.001[a] |
| Related to medications | 36 | 21.4 | 12 | 21.8 | 2 | 4.7 | 50 | 18.8 | |
| Related to nosocomial infection | 15 | 8.9 | 10 | 18.2 | 10 | 23.3 | 35 | 13.2 | |
| Related to procedures | 58 | 34.5 | 23 | 41.8 | 27 | 62.8 | 108 | 40.6 | |
| Related to diagnoses | 0 | 0.0 | 2 | 3.6 | 0 | 0.0 | 2 | 0.8 | |
| Other types of AE | 7 | 4.2 | 2 | 3.6 | 2 | 4.7 | 11 | 4.1 | |
| **Preventability of AE** | | | | | | | | | |
| Non-preventable | 77 | 45.8 | 19 | 34.5 | 15 | 34.9 | 111 | 41.7 | 0.206[b] |
| Preventable | 91 | 54.2 | 36 | 65.5 | 28 | 65.1 | 155 | 58.3 | |
| **Comorbidity** | | | | | | | | | |
| No | 3 | 1.8 | 0 | 0.0 | 0 | 0.0 | 3 | 1.1 | 1.000[a] |
| Yes | 165 | 98.2 | 55 | 100.0 | 43 | 100.0 | 263 | 98.9 | |
| **Hospitalization sector** | | | | | | | | | |
| Clinical | 76 | 45.2 | 19 | 34.5 | 6 | 14.0 | 101 | 38.0 | **0.002[a]** |
| Surgical | 82 | 48.8 | 32 | 58.2 | 34 | 79.1 | 148 | 55.6 | |
| Intensive / semi-intensive care | 10 | 6.0 | 4 | 7.3 | 3 | 7.0 | 17 | 6.4 | |
| **Total** | 168 | 100 | 55 | 100 | 43 | 100 | 266 | 100 | - |

AE, adverse event.

p-values were calculated using

[a]Fisher's exact test and

[b]Pearson's chi-square test;

Significant numbers from the statistical tests are presented in bold.

**Table 4. Association between the preventability of adverse events and classification of the events according to their nature.**

| Variable | Preventability of AE | | | | Total | | p-value[a] |
|---|---|---|---|---|---|---|---|
| | Non-preventable | | Preventable | | | | |
| | n | % | n | % | n | % | |
| **Classification of AE** | | | | | | | |
| Related to general care | 15 | 13.5 | 45 | 29.0 | 60 | 22.6 | **0.031** |
| Related to medications | 24 | 21.6 | 26 | 16.8 | 50 | 18.8 | |
| Related to nosocomial infection | 15 | 13.5 | 20 | 12.9 | 35 | 13.2 | |
| Related to procedures | 53 | 47.7 | 55 | 35.5 | 108 | 40.6 | |
| Related to diagnoses | 0 | 0.0 | 2 | 1.3 | 2 | 0.8 | |
| Other types of AE | 4 | 3.6 | 7 | 4.5 | 11 | 4.1 | |
| **Total** | 111 | 100 | 155 | 100 | 266 | 100 | - |

AE, adverse event.

[a] p-value was calculated using Fisher's exact test; Significant number from the statistical test is presented in bold.

**Table 5. Risk factors for the occurrence of adverse events.**

| | Adverse event | | RR (95% CI) |
|---|---|---|---|
| | **No** | **No** | |
| | **n (%)** | **n (%)** | |
| **Sex** | | | |
| Female | 15 (19.5) | 62 (80.5) | 0.94 (0.81–1.08) |
| Male | 10 (13.9) | 62 (86.1) | 1.00 |
| **Length of hospital stay** | | | |
| 1 to 7 days | 19 (24.1) | 60 (75.9) | 1.00 |
| 8 days or more | 6 (8.6) | 64 (91.4) | 1.20 (1.04–1.39) |
| **Age group** | | | |
| <60 years old | 13 (16.7) | 65 (83.3) | 1.00 (0.87–1.16) |
| 60 years or older | 12 (16.9) | 59 (83.1) | 1.00 |
| **Hospital discharge** | | | |
| Discharged alive | 23 (16.9) | 113 (83.1) | 0.98 (0.77–1.25) |
| Died | 2 (15.4) | 11 (84.6) | 1.00 |
| **Hospital admission** | | | |
| Elective | 21 (16.8) | 104 (83.2) | 0.99 (0.82–1.21) |
| Urgency | 4 (16.7) | 20 (83.3) | 1.00 |
| **Education** | | | |
| None | 2 (14.3) | 12 (85.7) | 0.94 (0.71–1.25) |
| Incomplete elementary school | 3 (12.5) | 21 (87.5) | 0.96 (0.76–1.22) |
| Complete elementary education | 14 (19.7) | 57 (80.3) | 0.88 (0.71–1.10) |
| Incomplete high school | 2 (40.0) | 3 (60.0) | 0.66 (0.32–1.38) |
| Complete high school | 3 (12.5) | 21 (87.5) | 0.96 (0.76–1.22) |
| Complete higher education | 1 (9.1) | 10 (90.9) | 1.00 |
| **Intrinsic factors** | | | |
| No | 3 (15.0) | 17 (85.0) | 1.02 (0.84–1.25) |
| Yes | 22 (17.1) | 107 (82.9) | 1.00 |
| **Extrinsic factors** | | | |
| No | 1 (25.0) | 3 (75.0) | 0.90 (0.51–1.59) |
| Yes | 24 (16.5) | 121 (83.5) | 1.00 |
| **Comorbidity** | | | |
| No | 1 (25.0) | 3 (75.0) | 0.90 (0.51–1.59) |
| Yes | 24 (16.5) | 121 (83.5) | 1.00 |
| **Procedure** | | | |
| No | 10 (23.3) | 33 (76.7) | 0.89 (0.75–1.07) |
| Yes | 15 (14.1) | 91 (85.9) | 1.00 |

RR, relative risk; CI, confidence interval.

It is assumed that the space-time attribute of this research, when confronted with others of the same methodological design, may have influenced the determination of AE cases, since the concept of patient safety has evolved continuously as have the parameters for identifying safety risks for hospital patients. The risks faced by patients in the health care process have also increased over time, and health care has been converted into an extremely complex system, since it has incorporated more effective and invasive diagnostic and therapeutic procedures. In addition, patients with a higher burden of disease and who are using several different medications are currently being admitted, requiring the performance of a multidisciplinary health

team. Such elements act to increase the probability of AE occurrence during the provision of care [18, 33].

With regard to the demographic variables of the patients in this study, it was noted that these characteristics did not represent risk factors for the occurrence of AEs. A similar study conducted at a university hospital in Tunisia concluded that the data corresponding to the age and sex of patients with AEs did not differ from those found in patients without AEs [25]. Other studies carried out in England, Spain, Italy, Ireland, Iran, and Portugal also found that there was no significant difference in gender between the two groups. However, these same studies stated that advancing age is a risk factor for the occurrence of AEs, given that the incidence of AEs in elderly patients was higher than that in younger patients [9, 18, 20, 28, 34–36].

In line with this study, in Tunisian research the intrinsic risk factors did not show a relationship with the occurrence of AEs; however, the rate of AEs was significantly higher in patients who had extrinsic risk factors [25]. A Chilean study found that, among patients who suffered AEs, 58.1% had intrinsic risk factors, and the most recurrent were: arterial hypertension, diabetes, hypoalbuminemia, and obesity; while 96.8% of the subjects exhibited extrinsic risk factors, for example: the presence of a peripheral venous catheter, use of a continuous infusion pump, and the presence of an indwelling urinary catheter [19]. In this scenario, it is essential to emphasize the trend of the dose-response effect indicated by a Spanish study, wherein the increase in the number of extrinsic and intrinsic factors per patient was accompanied by an increase in the number of AEs [20]. Specifically regarding extrinsic factors, it is noted that they are directly related to the care provided, suggesting the need to improve the quality of care processes in health services and proving the indispensability of professional competence, both essential components for the prevention of AEs.

Among the variables related to hospitalization aspects, this study established a relationship between the length of hospital stay and the occurrence of AEs, showing that the length of hospital stay represented a risk factor. The risk of occurrence of AEs in patients hospitalized for more than a week was 1.2 times greater than the risk for those whose stay was less than one week. Other studies have also associated the length of hospital stay with the occurrence of AEs, such as the case of the retrospective review of medical records carried out in hospitals in eight countries with developing or transitional economies, in 2005, which found that the rate of AEs increased with the length of stay, starting at 4% and expanding to 25% for periods of approximately 30 days [17].

Iranian researchers have found that an increase of one day in the length of stay may increase the possibility of AEs by 6.6% [35]. Another study carried out in Dutch hospitals in the years 2011 and 2012 discussed the probable reasons why patients with a longer hospital stay exhibited a higher incidence of AEs: patients with longer hospital stays receive interventions more frequently and are more likely to suffer from AEs; from another perspective, the patient who develops an AE during his hospitalization may have to stay in hospital for longer [37]. Such information justifies the implementation and consolidation of patient safety interventions, which aim to improve the care provided and reduce avoidable costs for the health system.

The most common types of AEs observed in this study were related to procedures (40.6%), especially those related to surgery. Thereafter, the most common types of AEs in order of frequency of occurrence were events related to general care such as pressure injuries, phlebitis, and falls (22.6%); medications (18.8%); nosocomial infection (13.2%); any other type of AE not specified such as unavailability of inputs and allergic reactions unrelated to the medication or of unknown origin (4.1%); and AEs related to delayed diagnosis (0.8%). Research carried out in the Catalonia region of Spain showed that AEs associated with surgical intervention were the most frequent (38.2% out of 356 AEs), followed by those related to nosocomial infection (22.8%), invasive non-surgical procedures (18.8%), and relating to the use of medicines

(17.7%) [38]. A Brazilian study showed that the most frequent types of AEs were related to surgical procedures (35.2%), clinical procedures (30.6%), diagnoses (10.2%), and medications (5.6%) [15].

In this study, among 155 AEs (58.3%) considered to be preventable, 35.5% were related to procedures, 29.0% to general care, 16.8% to medications, 12.9% to nosocomial infection, and 1.3% to diagnoses. It is emphasized that a statistically significant difference was observed between the types of AEs and their preventability.

A systematic review and meta-analysis found that among the 70 studies included, 60% reported preventable AEs related to the use of drugs, non-drug therapeutic management, diagnoses, non-surgical invasive procedures, surgical procedures, and infections acquired during health care. This review was based on a sample of 47,148 AEs, of which 25,977 (55%) were preventable [39], showing a proportion similar to that estimated in this study.

Regarding the preventable AE rates found in other worldwide surveys, the following occurrence percentages were obtained: USA (27.6%), Australia (51.2%), New Zealand (37.1%), United Kingdom (48%), Canada (36.9%), Denmark (40.4%), France (27.7%), Sweden (70%), Spain (42.8%), Tunisia (60%), Netherlands (39.6%), Italy (56.7%), Portugal (53.2%), Colombia (61%), Mediterranean countries (83%), Latin American countries (60%), Ireland (72.7%), Iran (34.3%), and Switzerland (49%) [5–10, 15–18, 21–29, 31, 40].

The assessment of preventability is a major challenge, given the need for an individual analysis of cases for decision making. Although retrospective review studies use criteria, concepts, and standards, the classification of preventable AEs includes a subjective element, which may vary according to the reviewer's experience and the way the information is documented in the patients' medical records. In addition, knowledge about the outcomes and severity of AEs can influence the judgment of preventability, causing a retrospective bias and creating a trend that is likely to overestimate the rate of preventable AEs [28].

In addition, regarding the preventability of AEs, this study found that there was no association between the preventable and non-preventable AE groups and their severity, confirming the results of the Canadian and Spanish studies, which showed that preventability is independent of the severity of AEs [6, 16].

In opposition to the relationship between severity and preventability, this research found a statistically significant association between the categories of severity and the types of AEs as well as with the hospitalization sectors. The surgical sector represented the location with the highest frequency of AEs in all severity categories. Procedure-related AEs had the highest number of occurrences at all severity levels. In addition, among the mild events, AEs related to general care and medications were highlighted; among AEs of moderate severity, those related to medications were relevant; and among the severe events, the importance of AEs related to hospital-acquired infection was perceived. Regarding the impact of AEs, data on the severity levels disseminated by other retrospective medical record studies reported conditions similar to those found in this study. The Spanish study showed that 45% of AEs were considered mild, whereas 38.9% were moderate, and 16% were severe. In a study carried out in a private hospital in Chile, the authors pointed out that 78.4% of AEs were categorized as mild, 18.9% as moderate, and 2.7% as severe [16, 19].

Therefore, AEs referring to procedures, medications, general care, infection related to health care, diagnoses, and other types of AEs, must represent priority areas of action aimed at mitigating the harm to patients, especially those that are preventable. The results of preventability and severity of these AEs are reflected in recommendations from international initiatives for patient safety policy.

This study was limited because the retrospective analysis of clinical records proved to be a complex method of operation and application for broader monitoring, due to the associated

costs, limitations of access to the records of some patients, dependence on the information documented in the medical records, demand for researchers with availability and specific clinical knowledge, and reasonable reliability in the reviewer's judgment, since the reliability of the review process represents a critical element in this type of study.

## Conclusions

The incidence of AEs related to health care was 33.7%. AEs were classified, in order of occurrence, as being related to procedures, general care, and medications. Regarding the severity of AEs, it was found that 63.2% were mild, 20.7% were moderate, and 16.2% were severe. Regarding the preventability of AEs, it was estimated that 58.3% were preventable. The length of a patient's hospital stay was identified as a risk factor for the occurrence of AEs.

Through knowledge of the incidence, nature, severity, preventability, and risk factors for the occurrence of AEs in the hospital environment, this study provides an overview of the types of problems that permeate patient safety. Obtaining local evidence on the complexity of the problem and its clinical impact creates the opportunity to prioritize the implementation of strategies for mitigating specific AEs based on reliable data and concrete information, in addition to promoting a culture of patient safety in health institutions.

A wider dissemination of studies using the retrospective medical record review method is necessary, since a global and current analysis is extremely important, because as pointed out, previous studies are not recent and patient safety has advanced a lot, as well as the safety culture, facts that promote reports in medical records and notifications by health professionals. A worldwide analysis is essential in order to question the results of these studies and make it possible to combine the retrospective medical record review method with an assessment of the patient safety culture or the implementation of quality management in hospitals, enabling any association between the incidence rate of AEs and the quality of care provided in hospital institutions.

## Acknowledgments

The authors would like to thank the research team of researcher Dr. Walter Viera Mendes Junior (in memoriam), and the researchers of the Canadian Adverse Event Study, G. Ross Baker and Virginia Flintoft, who collaborated in the development of this research.

## Author Contributions

**Conceptualization:** Ariane Cristina Barboza Zanetti, Rodrigo Soato, Carmen Silvia Gabriel.

**Data curation:** Ariane Cristina Barboza Zanetti, Bruna Moreno Dias, André Almeida de Moura, Rodrigo Soato, Carmen Silvia Gabriel.

**Formal analysis:** Ariane Cristina Barboza Zanetti, Bruna Moreno Dias, Carmen Silvia Gabriel.

**Investigation:** Ariane Cristina Barboza Zanetti, Bruna Moreno Dias, André Almeida de Moura, Carmen Silvia Gabriel.

**Methodology:** Ariane Cristina Barboza Zanetti, Bruna Moreno Dias, Andrea Bernardes, Helaine Carneiro Capucho, Alexandre Pazetto Balsanelli, Rodrigo Soato, Carmen Silvia Gabriel.

**Project administration:** Ariane Cristina Barboza Zanetti, Carmen Silvia Gabriel.

**Software:** Ariane Cristina Barboza Zanetti.

**Supervision:** Ariane Cristina Barboza Zanetti, Carmen Silvia Gabriel.

**Writing – original draft:** Ariane Cristina Barboza Zanetti, Bruna Moreno Dias, Andrea Bernardes, Helaine Carneiro Capucho, Alexandre Pazetto Balsanelli, Carmen Silvia Gabriel.

**Writing – review & editing:** Ariane Cristina Barboza Zanetti, Bruna Moreno Dias, Andrea Bernardes, Helaine Carneiro Capucho, Alexandre Pazetto Balsanelli, André Almeida de Moura, Rodrigo Soato, Carmen Silvia Gabriel.

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
