## [Decision Letter · Decision Letter 0]

22 Mar 2021

Incidence and preventability of adverse events in adult patients admitted to a Brazilian teaching hospital

PONE-D-20-36899

Dear Dr. Zanetti,

We’re pleased to inform you that your manuscript has been judged scientifically suitable for publication and will be formally accepted for publication once it meets all outstanding technical requirements.

Kind regards,

Ravishankar Jayadevappa

Academic Editor

PLOS ONE

Journal Requirements:

1. Please include in your Methods section (or in Supplementary Information files) the participating hospitals/institutions.

Furthermore, please provide details or citation for the six level scale used in the study [line 145]. Please clarify whether the scale has been previously validated?

Reviewers' comments:

Reviewer's Responses to Questions

**Comments to the Author**

1. Is the manuscript technically sound, and do the data support the conclusions?

Reviewer #1: Yes

Reviewer #2: Yes

2. Has the statistical analysis been performed appropriately and rigorously? 

Reviewer #1: Yes

Reviewer #2: Yes

3. Have the authors made all data underlying the findings in their manuscript fully available?

Reviewer #1: Yes

Reviewer #2: Yes

4. Is the manuscript presented in an intelligible fashion and written in standard English?

Reviewer #1: Yes

Reviewer #2: Yes

5. Review Comments to the Author

Reviewer #1: The objective of this study is to analyze the incidence and preventability of adverse events related to health care in adult patients admitted to a Brazilian teaching hospital. This is an interesting manuscript that focuses on an issue that is commonly encountered in clinical car. The manuscript is well written, clearly presented and easy to follow.

Reviewer #2: Incidence and preventability of adverse events in adult patients admitted to a Brazilian teaching hospital

Variations in quality of care is a ubiquitous feature of the healthcare systems across world. Healthcare providers’ contribution to the observed variation in quality of care across region, hospital, age, and racial group is an important and unexplored area in the health outcomes research. The objective of this retrospective study was ‘to analyze the incidence and preventability of adverse events related to healthcare in adult patients admitted to a Brazilian teaching hospital. Overall, this is a well done study and the authors must be congratulated for this important contribution to the existing literature on health services and outcomes research.

6. PLOS authors have the option to publish the peer review history of their article (what does this mean?). If published, this will include your full peer review and any attached files.

Reviewer #1: No

Reviewer #2: Yes: Ravishankar Jayadevappa.

---

## [Editor Report · Acceptance letter]

5 Apr 2021

PONE-D-20-36899 

Incidence and preventability of adverse events in adult patients admitted to a Brazilian teaching hospital 

Dear Dr. Zanetti:

I'm pleased to inform you that your manuscript has been deemed suitable for publication in PLOS ONE. Congratulations! Your manuscript is now with our production department. 

Kind regards, 

on behalf of

Dr. Ravishankar Jayadevappa 

Academic Editor

PLOS ONE